

# Comparison of survival in patients with low *vs.* intermediate prostate-specific antigen concentrations and development of a nomogram: a surveillance, epidemiology and end results program database study with external validation on a Chinese cohort

Jingchang Mei, Guanqun Zhu, Yu Yao, Fengju Guan, Lijiang Sun and Guiming Zhang

Department of Urology, The Affiliated Hospital of Qingdao University, Qingdao, China

## ABSTRACT

**Background:** In this study of patients with prostate cancer, we explored associations between low prostate-specific antigen (PSA) concentrations and disease progression as well as prognosis.

**Methods:** We retrospectively reviewed data of 233,554 prostate cancer patients in the Surveillance, Epidemiology and End Results (SEER) program and of 199 prostate cancer patients from the medical records of the Affiliated Hospital of Qingdao University with PSA ≤10 ng/mL at diagnosis. The patients were stratified into eight subgroups by T stage and Gleason score (GS) and survival curves for the resultant subgroups plotted using the Kaplan–Meier method. Multivariate Cox analyses were performed to investigate the effects of PSA concentrations in different subgroups. After randomly dividing patients into a training set and an internal validation set with a ratio of 7:3, a nomogram model to predict the survival of prostate cancer patients was subsequently established and validated.

**Results:** In all prostate cancer patients with Gleason score (GS) 8–10, low PSA concentrations were significantly associated with advanced disease and poor prognosis, functioning as a statistically significant risk factor. Conversely, in patients with GS 6–7 and Stage T1 disease, low PSA concentrations acted as a protective factor. A nomogram model for predicting prognosis was established and validated. We obtained similar results with an external validation cohort.

**Conclusions:** Our findings indicate that low PSA concentrations exert divergent impacts on prostate cancer patients stratified by T stage and GS. Specifically, in patients with high GS (8–10), low PSA concentrations represent a risk factor for disease progression to advanced stages and poor prognosis. Additionally, we developed a novel nomogram that effectively predicts survival outcomes in these patients.

Corresponding author
Guiming Zhang,
zhangguiming9@126.com

## INTRODUCTION

Prostate cancer ranks as the second most common malignancy among men globally, posing a severe threat to patients' life and health (*Bergengren et al., 2023*). High prostate-specific antigen (PSA) concentrations are commonly thought to be associated with adverse clinical features and poor overall survival (OS). Nevertheless, some researchers have found that the association between PSA level and prognosis may not be linear; patients with low PSA concentrations (PSA ≤ 4 ng/mL) have been reported to have more aggressive prostate cancer, a worse prognosis than patients with intermediate PSA concentrations (PSA 4–10 ng/mL), and to be prone to achieve unsatisfactory results with conventional local treatment (LT), including radical prostatectomy (RP) and radiotherapy (RT) (*Gosselaar, Roobol & Schröder, 2005*; *Schröder et al., 2000*; *Wilt, 2014*). The characteristics of neuroendocrine differentiation in the population with low PSA concentrations may be related to it, especially in patients with a high Gleason score, but up to now, there is still no explanation that is credible enough (*Aggarwal et al., 2021*; *Ferraro, Rossi & Biganzoli, 2021*; *Mahal et al., 2018*).

Although some research groups have mentioned this phenomenon, to the best of our knowledge none have identified which categories of patients have these associations with low PSA concentrations. Using patients with slightly higher PSA concentrations (PSA 4–10 ng/mL) as a reference, we sought to analyze whether all patients with low PSA concentrations have poor prognoses or whether this is true only of patients with specific T stages or Gleason scores (GS). Furthermore, after identifying the affected patients, we tried to construct and validate a prognostic prediction model incorporating PSA concentration, LT, and other clinical variables.

## METHODS

### Study cohorts

The Surveillance, Epidemiology and End Results (SEER) program collects and reports data on diagnosis and treatment from population-based cancer registries and covers approximately 28% of the population of the USA. In this study, we used SEER*Stat 8.4.0 software to identify prostate cancer patients with PSA ≤ 10 ng/mL at diagnosis between 2010 and 2017 in the SEER 18 registry database. We extracted the following independent variables from the SEER database: GS, TNM-stage, PSA concentration at diagnosis, age, race, and therapy received (including RP and RT). Based on previous research, we selected GS 8–10 and GS 6–7 as the criteria for binary classification of GS (*Fankhauser et al., 2023*; *Mahal et al., 2018*). High-grade PCa was defined as a pre-treatment Gleason score of 8 or above and low/intermediate grade as a Gleason score of 6 or 7. Patients were stratified by the GS identified at biopsy into low/intermediate-grade and high-grade group. Individuals missing any of the information listed above were excluded. This selection process yielded 233,554 study patients. The primary study endpoint was OS (death for any reason).

An independent cohort was used for external validation. This external validation set comprised 199 prostate cancer patients with pre-treatment PSA concentrations ≤10 ng/mL
who were treated from May 2013 to September 2023 in the Affiliated Hospital of Qingdao University. The following variables were collected: age at diagnosis, TNM stage, GS score, vital status, and survival in months. This study was performed in line with the principles of the Declaration of Helsinki. Approved was granted by the medical ethics committee of the Affiliated Hospital of Qingdao University (QYFY WZLL 28372). All individual participants included in the study informed and consented to the possible use of relevant health data for scientific research.

## Statistical analysis

Baseline characteristics were compared between PSA concentration groups (≤4.0 ng/mL, 4.0–10.0 ng/mL), GS groups (6–7, 8–10), and T stage groups (T1, T2, T3 and T4). Continuous variables are expressed as medians and interquartile ranges and categorical variables as frequencies and proportions. Continuous variables were compared using the Mann–Whitney U test and categorical variables using the chi-square test. Patients were stratified into eight subgroups according T stage groups (T1, T2, T3 and T4) and GS groups (6–7, 8–10). Kaplan–Meier survival curves were constructed to compare the outcomes of different PSA concentration in eight subgroups. In Cox regression, hazard ratios > 1 was regarded as a risk factor, while hazard ratios < 1 was considered a protective factor. Univariate and multivariable Cox regression analyses were used to evaluate whether lower PSA concentrations were statistically significant risk factors for worse prognosis in these subgroups. Based on the results obtained, eight subgroups were then combined into high-risk group and low-risk group. Patients in high-risk group were randomly divided into a training set and an internal validation set in a 7:3 ratio for the construction and verification of the nomogram. The patients in the training set were then subjected to univariate and multivariate Cox regression analysis to define hazard ratios by PSA concentration. A nomogram model based on the results of these calculations in training set was established to predict 2-, 3- and 5-year OS for patients at high-risk. Receiver operating characteristic curves (ROC) were plotted based on the nomogram model, and its ability to discriminate tested using the area under the ROC curve (AUC). The calibration curves were used to internally validate the nomogram model and decision curve analysis to determine the net benefit of using this model. The external validation set from the Affiliated Hospital of Qingdao University was stratified into high- and low-risk groups based on the grouping criteria obtained through assessing the data from the SEER program. Again, continuous and categorical variables were analyzed using the Mann–Whitney U and chi-square tests, respectively. Kaplan–Meier survival curves for high- and low-risk groups were plotted and the log-rank test results calculated to determine the statistical significance of the findings. ROC curves were plotted and AUCs calculated to validate the obtained model. Statistical analyses were performed using Stata version 18.0 and R software version 4.3.0. All *P*-values were two-sided and $P < 0.05$ was considered to denote statistical significance.

## RESULTS

### Baseline characteristics

The median duration of follow-up was 63 months. As shown in Table 1, the 233,554 study patients were divided into two groups: GS 6–7 ($N$ = 203,038) and GS 8–10 ($N$ = 30,516). Patients in the GS 8–10 group with PSA ≤ 4 ng/mL had a significantly higher chance of nodal metastasis ($P$ < 0.001), distant metastasis ($P$ < 0.001), and advanced T stage disease ($P$ < 0.001). Next, patient in the two groups were stratified into four subgroups by T stage: T1–T4. Ultimately, based on different T stages and GS, a total of eight subgroups were created to observe the impact of different PSA concentration across the various subgroups (Tables S1 and S2).

### Survival outcomes

Kaplan–Meier survival curves revealed that patients in the GS 6–7 group with T1 prostate cancer and PSA ≤ 4 ng/mL had better prognoses than did those in the PSA 4.1–10 ng/mL subgroup (Fig. 1A). However, this was not the case for the T2, T3 and T4 subgroups with GS 6–7 (Figs. 1B–1D). In the GS 8–10 group, PSA ≤ 4 ng/mL served as a risk factor for poor OS across all subgroups, in contrast to the finding in the T1 subgroup of patients with GS 6–7 (Figs. 1E–1H). The outcomes of multivariate Cox regression analyses according to T-category in the GS 6–7 and GS 8–10 groups are presented in Tables S3 and S4. PSA ≤ 4 ng/mL emerged as a risk factor across all T categories: T1 (HR, 1.18; 95% CI [1.04–1.35]; $P$ = 0.0136), T2 (HR, 1.20; 95% CI [1.07–1.35]; $P$ = 0.0026), T3 (HR, 1.20; 95% CI [1.01–1.42]; $P$ = 0.0331), and T4 (HR, 1.35; 95% CI [1.03–1.78]; $P$ = 0.0294). In contrast, in the GS 6–7 group, only patients with T1 stage prostate cancer and PSA ≤ 4 ng/mL were found to have a better survival (HR, 0.91; 95% CI [0.85–0.97]; $P$ = 0.005). There were no significant correlations between PSA ≤ 4 ng/mL and survival in the T2, T3 and T4 subgroups.

### Construction and validation of a nomogram model

Based on the association between PSA ≤ 4 ng/mL and prognosis identified by multivariate Cox regression analyses in above eight subgroups, we created low-risk (T1 and T2–T4 with GS 6–7) and high-risk (T1–T4 with GS 8–10) subgroups. Given the specific significance of low PSA concentrations in the high-risk group, we tried to develop a predictive model for survival in high-risk group patients with low PSA concentrations. After randomly assigning patients in high-risk group into the training set ($N$ = 21,361) and the internal validation set ($N$ = 9,155), univariate and multivariate Cox regression analysis were performed in the training set to define hazard ratios by PSA concentration and other factors (Table 2). Ultimately, the following were identified as independent prognostic factors: age, race, PSA concentration, LT, T, N and M stage. To establish a practical method for predicting the survival probability of patients in the high-risk group, we generated a nomogram using the training set to evaluate the probability of 2-, 3-, and 5-year survival (Fig. 2), after which we plotted ROC curves, the AUCs of which were 0.788,

**Table 1 Baseline characteristics of patients in SEER program.**

| Variable | Overall | PSA ≤ 4.0 ng/mL | PSA 4–10 ng/mL | P value |
|---|---|---|---|---|
| *N*, (%) | 233,554 | 39,536 (17.0) | 194,018 (83.0) | |
| Age, *N* (%) | | | | <0.001 |
| <65 | 112,755 (48.28) | 21,327 (53.94) | 91,428 (47.12) | |
| 65–69 | 58,343 (24.98) | 8,390 (21.22) | 49,953 (25.75) | |
| 70–74 | 37,031 (15.86) | 5,675 (14.35) | 31,356 (16.16) | |
| 75–79 | 18,015 (7.71) | 2,774 (7.02) | 15,241 (7.86) | |
| 80–84 | 5,806 (2.49) | 985 (2.49) | 4,821 (2.48) | |
| ≥85 | 1,604 (0.69) | 385 (0.97) | 1,219 (0.63) | |
| Race, *N* (%) | | | | <0.001 |
| White | 183,579 (78.60) | 32,515 (82.24) | 151,064 (77.86) | |
| Black | 34,795 (14.90) | 4,983 (12.60) | 29,812 (15.37) | |
| Other | 15,180 (6.50) | 2,038 (5.15) | 13,142 (6.77) | |
| Gleason score, *N* (%) | | | | <0.001 |
| 6 | 104,702 (44.83) | 21,551 (54.51) | 83,151 (42.86) | |
| 7 | 98,336 (42.10) | 13,796 (34.89) | 84,540 (43.57) | |
| 8 | 18,773 (8.04) | 2,338 (5.91) | 16,435 (8.47) | |
| 9 | 10,733 (4.60) | 1,622 (4.10) | 9,111 (4.70) | |
| 10 | 1,010 (0.43) | 229 (0.58) | 781 (0.40) | |
| Stage T, *N* (%) | | | | <0.001 |
| T1 | 102,607 (43.93) | 14,129 (35.74) | 88,478 (45.60) | |
| T2 | 104,717 (44.84) | 21,816 (55.18) | 82,901 (42.73) | |
| T3 | 25,550 (10.94) | 3,392 (8.58) | 22,158 (11.42) | |
| T4 | 680 (0.29) | 199 (0.50) | 481 (0.25) | |
| Stage N, *N* (%) | | | | 0.0038 |
| N0 | 230,031 (98.49) | 39,004 (98.65) | 191,027 (98.46) | |
| N1 | 3,523 (1.51) | 532 (1.35) | 2,991 (1.54) | |
| Stage M, *N* (%) | | | | <0.001 |
| M0 | 231,927 (99.30) | 39,175 (99.09) | 192,752 (99.35) | |
| M1 | 1,627 (0.70) | 361 (0.91) | 1,266 (0.65) | |
| Local treatment, *N* (%) | | | | <0.001 |
| Yes | 166,983 (71.50) | 26,688 (67.50) | 140,295 (72.31) | |
| No | 66,571 (28.50) | 12,848 (32.50) | 53,723 (27.69) | |

0.763, 0.735 in training set, and 0.792, 0.789, 0.752, respectively (Fig. S1). Additionally, we constructed calibration curves to evaluate the predictive capacity of the model (Fig. S2). The calibration curve closest to 45° indicated that the model predicted upstaging well. As seen in Fig. S3, the decision curve analysis indicated that our nomogram outperformed default strategies in terms of net benefit for prostate cancer survival prediction, demonstrating its practical value. Taken together, it was clear that the nomogram model had good predictive power and that the identified independent predictors played critical roles in determining prognosis.

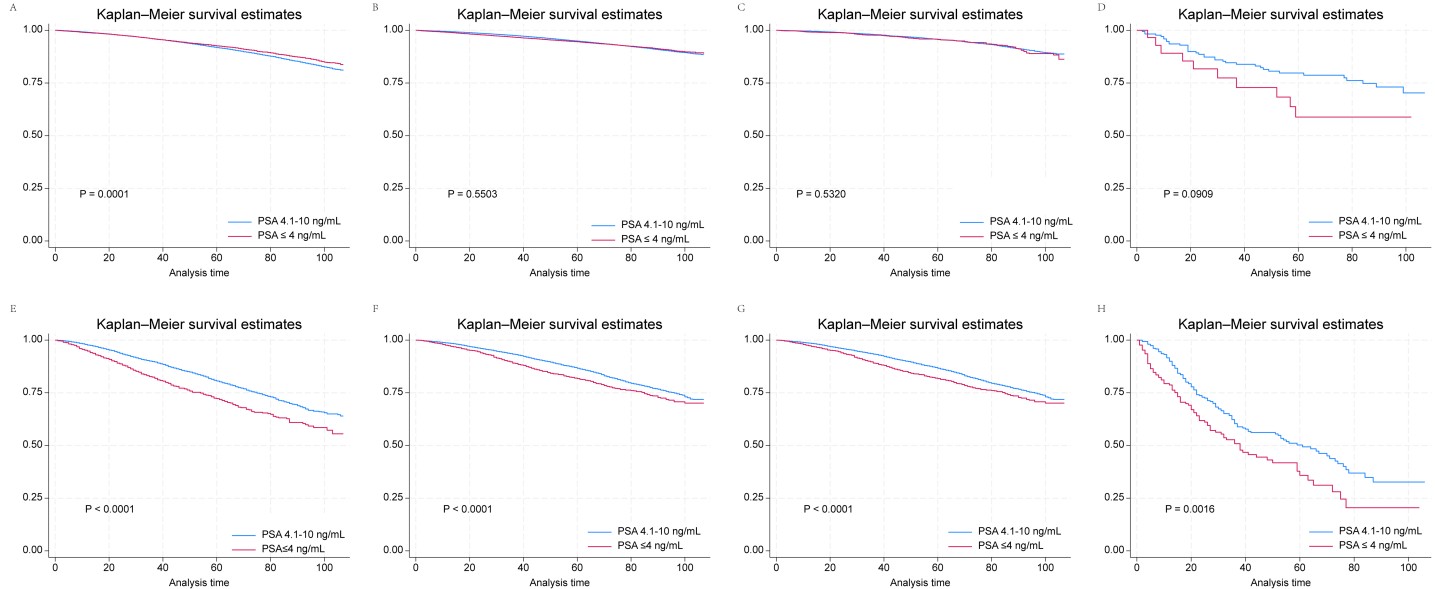

**Figure 1** Kaplan–Meier survival curves for patients with prostate cancer and PSA ≤ 4 ng/mL and 4.1–10.0 ng/mL according to their T stage and GS drawn from SEER data. (A) T1 and GS 6 and 7; (B) T2 and GS 6 and 7; (C) T3 and GS 6 and 7; (D) T4 and GS 6 and 7; (E) T1 and GS 8–10; (F) T2 and GS 8–10; (G) T3 and GS 8–10; (H) T4 and GS 8–10.

## External validation

The median duration of follow-up was 27 months. The external validation cohort consisted of 199 prostate cancer patients and was divided into low- (*N* = 130) and high-risk (*N* = 69) groups according to the grouping criteria developed above (Table 3). We found that men with PSA ≤ 4 ng/mL in the high-risk group had higher risks of nodal and distant metastasis. Similarly to the conclusions reached above, when we plotted Kaplan–Meier survival curves for the low- and high-risk groups (Fig. 3), we confirmed that PSA ≤ 4 ng/mL was significantly associated with worse prognosis in the high-risk group (*P* = 0.0336), whereas we identified no significant correlation in the low-risk group. ROC curves were plotted. For 2-year survival, the AUC was 95.5 (95% CI [88.6–100.0]); for 3-year survival, the AUC was 88.1 (95% CI [73.2–100.0]); and for 5-year survival, the AUC was 76.0 (95% CI [48.8–100.0]) (Fig. S4). Thus, the external validation cohort validated our model.

## DISCUSSION

PSA concentrations have been widely used to screen for prostate cancer and monitor disease progression. In general, PSA ≤ 4 ng/mL can be considered normal. However, some patients are diagnosed with prostate cancer despite having normal PSA concentrations (*Bonet et al., 2009*; *Izumi et al., 2015*; *Kang et al., 2020*; *Kobayashi et al., 2004*; *McGuire et al., 2012*). *Catalona, Smith & Ornstein (1997)* reported diagnosing prostate cancer in 22% of men with PSA concentrations between 2.6 and 4.0 ng/mL who underwent biopsy, and *Su et al. (2022)* reported diagnosing prostate cancer in 20.3% of men with PSA concentrations between 0 and 4.0 ng/mL. In patients diagnosed with prostate cancer, the

**Table 2 Univariate and multivariate Cox regression analyses of training set in SEER program.**

| Variable | Univariate Cox regression | | | Multivariate Cox regression | | |
|---|---|---|---|---|---|---|
| | HR | 95% CI | P value | HR | 95% CI | P value |
| Age | | | | | | |
| <65 | 1 | | | 1 | | |
| 65–69 | 1.30 | [1.16–1.45] | <0.001 | 1.32 | [1.18–1.47] | <0.0001 |
| 70–74 | 1.89 | [1.70–2.11] | <0.001 | 1.86 | [1.67–2.07] | <0.0001 |
| 75–79 | 2.52 | [2.26–2.82] | <0.001 | 2.34 | [2.08–2.62] | <0.0001 |
| 80–84 | 4.08 | [3.60–4.62] | <0.001 | 3.39 | [2.97–3.86] | <0.0001 |
| ≥85 | 9.92 | [8.58–11.48] | <0.001 | 5.99 | [5.12–6.99] | <0.0001 |
| Race | | | | | | |
| White | 1 | | | 1 | | |
| Black | 0.92 | [0.83–1.02] | 0.129 | 1.11 | [0.99–1.23] | 0.0622 |
| Other | 0.57 | [0.48–0.67] | <0.001 | 0.56 | [0.48–0.67] | <0.0001 |
| PSA level | | | | | | |
| 4.1–10.0 | 1 | | | 1 | | |
| ≤4.0 | 1.36 | [1.24–1.48] | <0.001 | 1.17 | [1.07–1.28] | 0.0009 |
| Stage T, N (%) | | | | | | |
| T1 | 1 | | | 1 | | |
| T2 | 0.69 | [0.64–0.75] | <0.001 | 0.85 | [0.78–0.92] | <0.0001 |
| T3 | 0.59 | [0.53–0.65] | <0.001 | 0.90 | [0.81–1.00] | 0.0314 |
| T4 | 4.10 | [3.46–4.85] | <0.001 | 2.53 | [2.12–3.03] | <0.0001 |
| Stage N, N (%) | | | | | | |
| N0 | 1 | | | 1 | | |
| N1 | 1.78 | [1.59–1.99] | <0.001 | 1.35 | [1.19–1.53] | <0.0001 |
| Stage M, N (%) | | | | | | |
| M0 | 1 | | | 1 | | |
| M1 | 7.09 | [6.40–7.86] | <0.001 | 3.60 | [3.20–4.05] | <0.0001 |
| Local treatment, N (%) | | | | | | |
| Yes | 1 | | | 1 | | |
| No | 3.78 | [3.52–4.07] | <0.001 | 2.12 | [1.95–2.31] | <0.0001 |

PSA concentration is commonly considered to correlate with disease progression and prognosis. A low PSA is typically seen as portending a favorable prognosis in patients with prostate cancer, some studies having provided evidence to support this position (*Antenor et al., 2005*). However, a contrary phenomenon has been observed; namely, that prostate cancer patients with low PSA concentrations (PSA ≤ 4 ng/mL) have worse prognoses than do those with the next highest PSA concentration (PSA 4–10 ng/mL). Several studies have found a J-shaped rather than a linear association between PSA and lethal disease among men with prostate cancer (*Fankhauser et al., 2020*; *Mahal et al., 2016*). As far as we know, whether this phenomenon applies to all patients with low PSA concentrations or only to some of them has not yet been clarified. To ascertain the significance of low PSA concentrations in patients with prostate cancer, in our study of patient data from the SEER

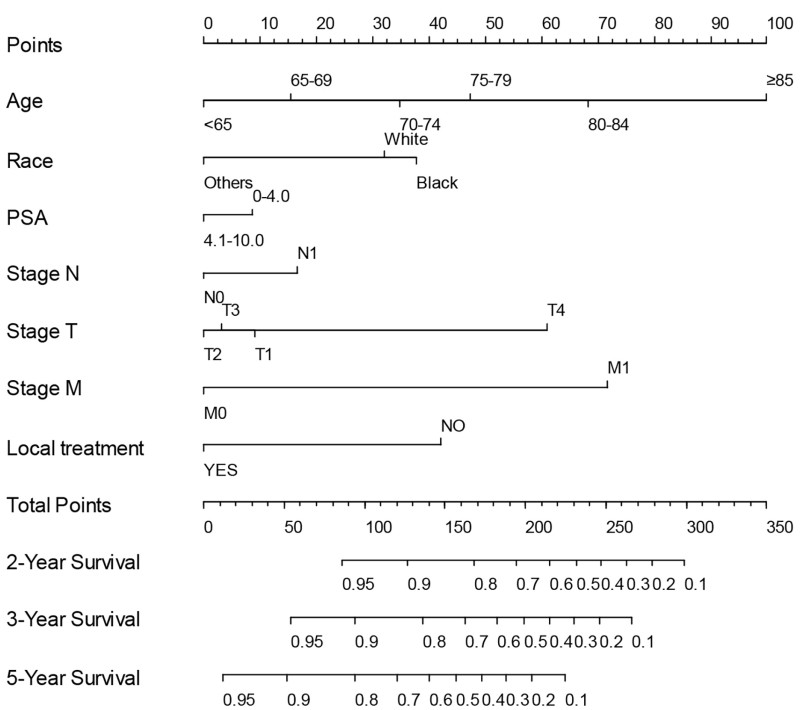

**Figure 2 Nomogram for predicting the 2-, 3- and 5-year OS of high-risk prostate cancer patients using training set from the SEER program.**

program we divided patients into various subgroups based on the T stage and Gleason score and analyzed our data according to those subgroups. These analyses revealed that the combination of low PSA concentration and GS ≥ 8 in patients with prostate cancer is uniquely associated with aggressive disease and a poor prognosis, PSA being a protective factor for survival only in patients with GS 6–7, T1 prostate cancer. We classified patients in whom low PSA is a risk factor into a high-risk group and those in whom low PSA is a protective factor or is not significantly associated with prognosis into a low-risk group. Our findings and those of previous studies suggest that as yet unidentified clinical and genetic characteristics may differ between these high- and low-risk groups (*Aggarwal et al., 2021*; *Antenor et al., 2005*; *Mahal et al., 2016*, *2018*).

In addition, to predicting the prognosis of prostate cancer patients in a high-risk group, a nomogram predicting 2-, 3- and 5-year OS was developed based on selected variables. ROC curves were used to evaluate the discrimination. Calibration curves were used to determine the degree of agreement between predicted probabilities and observed outcomes and decision curve analysis plots to estimate the clinical usefulness and benefits of the nomogram. Moreover, an external validation cohort comprising data of prostate cancer patients from the Affiliated Hospital of Qingdao University was used to verify the conclusions obtained from analyzing the SEER data.

Previous studies have mainly focused on associations between low PSA concentrations and disease progression and prognosis in prostate cancer patients with high GS (*Aggarwal et al., 2021*; *Mahal et al., 2016*, *2018*). There has been limited research on the significance of

**Table 3 Baseline characteristics of patients in the affiliated hospital of Qingdao University.**

| Variable | Overall | Low-risk group | High-risk group | P value |
|---|---|---|---|---|
| N, (%) | 199 | 130 (65.33) | 69 (34.67) | |
| Age, median (IQR) | 68.181 (7.084) | 67.792 (6.904) | 68.913 (7.406) | <0.001 |
| PSA, N (%) | | | | 0.0093 |
| PSA ≤ 4.0 ng/mL | 44 (22.11) | 21 (16.15) | 23 (33.33) | |
| PSA 4–10 ng/mL | 155 (77.89) | 109 (83.85) | 46 (66.67) | |
| Stage T, N (%) | | | | 0.0002 |
| T1 | 42 (21.11) | 33 (25.38) | 9 (13.04) | |
| T2 | 136 (68.34) | 92 (70.77) | 44 (63.77) | |
| T3 | 11 (5.53) | 3 (2.31) | 8 (11.59) | |
| T4 | 10 (5.03) | 2 (1.54) | 8 (11.59) | |
| Stage N, N (%) | | | | 0.0001 |
| N0 | 187 (93.97) | 129 (99.23) | 58 (84.06) | |
| N1 | 12 (6.03) | 1 (0.77) | 11 (15.94) | |
| Stage M, N (%) | | | | |
| M0 | 186 (93.47) | 130 (100.00) | 56 (81.16) | <0.0001 |
| M1 | 13 (6.53) | 0 (0.00) | 13 (18.84) | |
| Local treatment, N (%) | | | | 0.0001 |
| Yes | 178 (89.45) | 125 (96.15) | 53 (76.81) | |
| No | 21 (10.55) | 5 (3.85) | 16 (23.19) | |

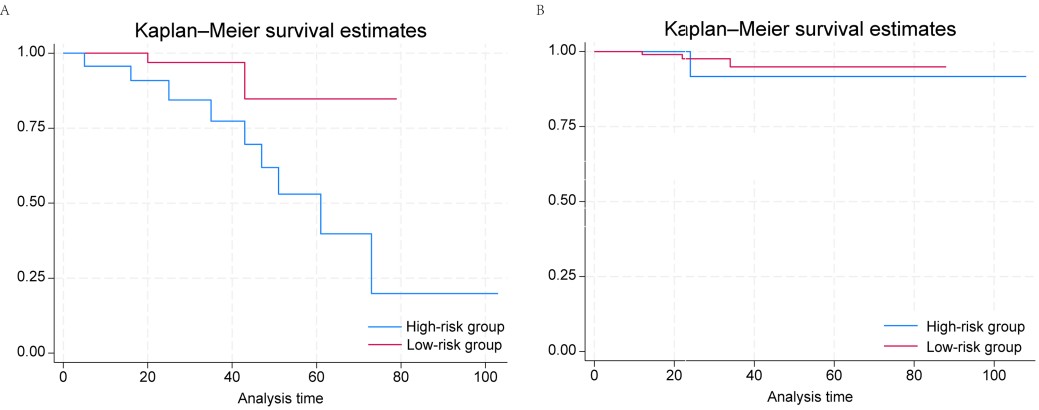

**Figure 3 Kaplan–Meier survival curves of PSA ≤ 4 ng/mL and 4.1–10.0 ng/mL groups in (A) low-risk and (B) high-risk prostate cancer patients from the affiliated hospital of Qingdao University.**

low PSA concentrations in patients with different T-stages of prostate cancer. *Antenor et al. (2005)* have reported that patients with T1c-stage prostate cancer and low PSA concentrations have a greater rate of organ-confined disease and better 10-year progression-free survival. This is contrary to the findings of previous studies on low PSA prostate cancer cited above, suggesting that T stage may be an important determinant of the prognosis of prostate cancer patients with low PSA concentrations. In our study,

T stage played a critical role. In the high-risk group, even though we found that low PSA concentrations are associated with poor prognosis for all T stages, the hazard ratio for low PSA concentrations in patients with T4 prostate cancer was significantly higher than for the other T stages. In the low-risk group, we found that low PSA concentration was a protective factor for prognosis in patients with T1 disease, and T stages above T1 were not associated with any significant differences in prognosis and PSA concentrations. Interestingly, Kaplan–Meier survival curves showed a non-statistically significant trend towards low PSA concentrations being associated with worse prognosis in low-risk patients with T4 stage prostate cancer, similar to the high-risk group. These findings suggest that as T stage and GS increase, low PSA concentrations gradually transition from being a protective to an increasingly important risk factor.

In agreement with some previous studies, our findings suggest that prostate cancer patients with low PSA concentrations may have some unique characteristics that result in low PSA concentrations having different effects on disease progression and prognosis with different T stages and GS. *Mahal et al. (2016)* proposed that low PSA concentrations with GS ≥ 8 may be an indicator of aggressive and very poorly differentiated or anaplastic low PSA–producing tumors, and provided evidence to support this contention. *Su et al. (2022)* reported finding several uncommon types of malignancies, including sarcoma, mucinous adenocarcinoma, neuroendocrine carcinoma, lymphoma, urothelial carcinoma, and solitary fibrous tumor, in patients with low PSA concentrations, which may explain why some patients with normal PSA concentrations have invasive prostate cancers. At the cellular level, prostate cancer cells that express little or no PSA have been found to play differential roles in tumor maintenance and progression. They demonstrate strong capacity for initiating robust tumor development and resisting androgen deprivation therapy (ADT) and may be a critical source of castration-resistant prostate cancer cells (*Qin et al., 2012*). Regarding genetic characteristics, *Mahal et al. (2018)* reported that tumors with low PSA concentrations and high GS had stronger expression of neuroendocrine/small-cell markers and decreased androgen receptor activity, whereas they found no such relationship for GS 7 tumors. This finding reveals possible differences between patients with low PSA concentrations, providing a potential explanation for the discrepancy between high- and low-risk groups in our study. *Conti et al. (2021)* have developed a polygenic risk score (PRS) for risk stratification that combines the 269 prostate cancer-associated genetic variants that account for 33.6–43.2% of prostate cancer familial relative risk. *Ma et al. (2023)* found that this PRS was associated with risk of lethal prostate cancer and that this association was stronger for those who had very low PSA concentrations. Unfortunately, these studies did not propose the possible genetic mechanisms for the characteristics of low PSA prostate cancer. *Aggarwal et al. (2021)* reported that, in patients with metastatic, castration-resistant prostate cancer, low PSA secretion was associated with metastatic tumor burden and loss of *RB1* and/or *TP53* was more common than in patients without these characteristics. In another study on the correlation between genomic characteristics and PSA concentration when screening, diagnosing, or monitoring prostate cancer, *KLK3* gene inactivation appeared to be associated with false-negative PSA findings (*Rodriguez et al., 2013*). Up to now, even

though some studies have explored the mechanisms underlying low PSA prostate cancer, there has been no consensus on the reasons for the unique characteristics of progression and prognosis exhibited by these prostate cancers. More work needs to be done with the aim of providing guidance for the diagnosis and treatment of this category of prostate cancer patients.

The current treatment strategies for localized prostate cancer mainly focus on LT, including RP and RT. Although there is no robust evidence for LT being advantageous in patients with low PSA concentrations, *Liu et al. (2020)* reported that LT achieves longer survival than does non-LT. *Guo et al. (2019)* found that, compared with external beam RT or external beam RT with brachytherapy, RP contributes to a significant increase in OS of patients with low PSA concentrations and high-grade prostate cancer, whereas external beam RT+ brachytherapy may be a better option than external beam RT alone in patients with very low PSA concentration. *Berglund et al. (2009)* surveyed 6,130 prostate cancer patients treated by RP and found no evidence that patients with low PSA concentrations had worse outcomes. According to those studies, LT, especially RP, does provide a satisfactory treatment benefit. Despite the absence of clear evidence, ADT resistance is thought to be a potential cause for poor prognosis in patients with low PSA prostate cancer (*Mahal et al., 2018*). Chemotherapy was proven to significantly prolong the overall survival of patients even in castration sensitive prostate cancers, while neither docetaxel nor cabazitaxel target androgen-AR signaling (*Amato, Stepankiw & Gonzales, 2013*; *Izumi et al., 2015*). This indicated the possibility of chemotherapy as a viable alternative form of treatment for patients with low PSA concentration to address possible ADT resistance, and some studies which suggested that addition of chemotherapy to standard hormonal therapy for localized disease may offer the greatest benefit to patients with low PSA concentrations and high grade disease have proven the potential of chemotherapy (*Sandler et al., 2015*; *James et al., 2016*; *Mahal et al., 2018*). In short, the treatment strategy for patients with low PSA needs to be supported by more research on LT, ADT, chemotherapy, and other factors. Currently, LT, especially RP, remains the preferred option for patients with localized prostate cancer.

Early screening for, and diagnosis of low PSA prostate cancer are urgently needed because of this condition's unique characteristics and prognosis. PSA concentration, generally an important biomarker in screening for prostate cancer, is ineffective and does not prompt a biopsy in individuals with prostate cancer and normal PSA concentrations. Recognized risk factors, including suspicious findings on digital rectal examination (DRE), a strong family history, and rapidly increasing PSA concentrations, are common tools in screening for and diagnosing prostate cancer with normal PSA concentrations. DRE, an easy-to-perform and cost-effective screening method, plays a crucial role in identifying patients with prostate cancer and low PSA concentrations. A significant proportion of low PSA prostate cancers are identified by DRE. Other possible biomarkers are being explored for use in screening for this disease. *Pelzer et al. (2005)* reported that the proportion of free to total serum PSA (f/tPSA ratio) can be useful when screening individuals with low PSA concentrations, whereas *Miyoshi, Kawahara & Uemura (2022)* found that serum

dehydroepiandrosterone concentration can help in predicting candidates for active surveillance prior to prostate biopsy. It is expected that more indicators will be discovered and verified. Some researchers have proposed a threshold PSA concentration of 2.5 ng/mL for recommending prostate biopsy. However, there is no clear evidence for a significant difference between patients with PSA 2.5–4 ng/mL and PSA ≤ 4 ng/mL (*Crawford et al., 2011*; *Gilbert et al., 2005*; *Kim et al., 2010*; *Meeks et al., 2009*; *Müntener et al., 2010*; *Su et al., 2022*). Further, lowering the threshold could lead to overtreatment. This proposal therefore requires further discussion and exploration. Moreover, other factors, including far lateral tumor location, tumor invasion, and microvessel density, may influence the PSA concentration, disease progression, and prognosis. More research is needed to determine the optimal means of screening for, and diagnosing these prostate cancers.

Our research had some limitations. First, because it was a retrospective study based on data from the SEER program and the Affiliated Hospital of Qingdao's records, there will inevitably be some confounding factors that could cause biases. Second, the SEER database lacked information about the treatment of prostate cancer patients with ADT and a lot of information on chemotherapy was missing, making these data difficult to analyze. Third, the number of patients in our clinical cohorts were relatively small compared to the SEER cohort. Despite this, we still identified significant differences in the high-risk group. Fourth, the present study is limited by the single-race nature of the external validation cohort, making it impossible to comprehensively evaluate the impact of racial and regional differences on the model. In the future, more diversified data need to be incorporated to enhance the model's universality. Finally, vigilance is required: biopsies are less likely to be performed on patients with normal PSA concentrations than on those with high PSA concentrations unless the former have identifiable risk factors, including suspicious findings on imaging or DRE, a strong family history, or a rapidly increasing PSA. The failure to perform a biopsy may delay the diagnosis of patients with normal PSA concentration, leading to more advanced disease and worse prognosis by the time of diagnosis, potentially causing biases.

Briefly, we recognize that, despite some controversies, high GS, low-PSA prostate cancer is generally considered to be associated with advanced disease and a poor prognosis. LT, especially RP, remains the preferred treatment strategy for this disease. Considering its unique oncologic outcomes, more research should be conducted on genetic characteristics, biomarkers, imaging information, novel therapeutic modalities, and clinical management of patients with high GS, low-PSA prostate concentrations cancer.

## CONCLUSION

Low PSA concentrations are associated with advanced disease and poor prognosis in all prostate cancer patients with GS 8–10, whereas they are a protective factor in those with GS 6–7 and T1 stage disease. We have established a nomogram model that could contribute to predicting the survival of patients with GS 8–10, low PSA concentrations prostate cancer.

## ACKNOWLEDGEMENTS

We thank Dr Trish Reynolds, MBBS, FRACP, from Liwen Bianji (Edanz) for editing the English text of a draft of this manuscript.

## ABBREVIATIONS

| | |
|---|---|
| **ADT** | androgen deprivation therapy |
| **DRE** | digital rectal examination |
| **GS** | Gleason score |
| **LT** | local treatment |
| **PRS** | polygenic risk score |
| **RP** | radical prostatectomy |
| **RT** | radiotherapy |

### Funding

This work was funded by the Natural Science Foundation of Shandong Province (ZR2021MH354). The funders had no role in study design, data collection and analysis, decision to publish, or preparation of the manuscript.

### Grant Disclosures

The following grant information was disclosed by the authors:
Natural Science Foundation of Shandong Province: ZR2021MH354.

### Competing Interests

The authors declare that they have no competing interests.

### Author Contributions

- Jingchang Mei conceived and designed the experiments, performed the experiments, analyzed the data, prepared figures and/or tables, authored or reviewed drafts of the article, and approved the final draft.
- Guanqun Zhu performed the experiments, prepared figures and/or tables, and approved the final draft.
- Yu Yao performed the experiments, authored or reviewed drafts of the article, and approved the final draft.
- Fengju Guan analyzed the data, authored or reviewed drafts of the article, and approved the final draft.
- Lijiang Sun analyzed the data, authored or reviewed drafts of the article, and approved the final draft.
- Guiming Zhang conceived and designed the experiments, prepared figures and/or tables, and approved the final draft.

## Human Ethics

The following information was supplied relating to ethical approvals (*i.e.*, approving body and any reference numbers):

The medical ethics committee of the Affiliated Hospital of Qingdao University approved the study (QYFY WZLL 28372).

## Data Availability

The raw data are available in the Supplemental Files.

## Supplemental Information

Supplemental information for this article can be found online at http://dx.doi.org/10.7717/peerj.19823#supplemental-information.

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
