# Peer review of "Comparison of survival in patients with low vs. intermediate prostate-specific antigen concentrations and development of a nomogram: a surveillance, epidemiology and end results program database study with external validation on a Chinese cohort"

_PeerJ, doi:10.7717/peerj.19823_

## Round 0.1 · original submission · Major Revisions

The authors are requested to carefully revise the manuscript and answer the questions raised by the reviewers.

Reviewer 1 ·

Basic reporting

The basic reporting is complete

Experimental design

The use of prediction and validation sets is very helpful, as is the addition of an external validation set

Validity of the findings

See additional comments below

Additional comments

I have a number of questions and comments:

1. Ordinarily, one would expect the AUC in the original prediction set to be higher than that in the validation set. However, this is not the case in Figure S1, ABC vs DEF. I wonder why not?

2. Please explain how to interpret the decision curve analysis in Figure S3 and lines 117 - 118.

3. It is great that an external validation set is provided. However, it is extremely small. Please, if possible, provide confidence intervals for the AUC estimates shown in Fig S4. Using cross-validation may help with this.

Reviewer 2 ·

Basic reporting

no comment

Experimental design

no comment

Validity of the findings

no comment

Additional comments

The article presents a retrospective cohort study investigating the prognostic impact of low versus intermediate prostate-specific antigen (PSA) concentrations in prostate cancer patients. The authors analyze survival outcomes using the SEER database and externally validate their findings with a Chinese hospital cohort. A predictive nomogram is developed to assess patient survival based on PSA levels, Gleason score (GS), T stage, and other clinical factors. However, there are some revisions required.
Major Issues
• The manuscript does not clearly explain how missing data were addressed. Since large databases like SEER often contain missing values, the authors should describe any imputation methods used or the impact of missing data on the analysis.
• The study accounts for key prognostic factors (e.g., Gleason score, T stage, PSA levels), but a deeper discussion on potential confounders (e.g., comorbidities, treatment variations, lifestyle factors) would strengthen the findings. Were adjustments made for these factors?
• While external validation was performed using a Chinese cohort, the applicability of the nomogram to diverse populations remains uncertain. The study should discuss whether ethnic and regional differences might influence the model’s predictive performance.
• The discussion could be improved by emphasizing how the findings should influence clinical decision-making. How should physicians incorporate the nomogram into current prostate cancer management strategies? Should PSA cutoffs be reconsidered based on these findings?
• While appropriate methods (Kaplan-Meier, Cox regression) are used, the manuscript should explicitly mention whether proportional hazards assumptions were tested in Cox regression analysis.
• Provide justification for choosing the 7:3 training-validation split, as different ratios might yield different model performance.
• The discussion acknowledges limitations, but the manuscript could explicitly mention selection bias (as patients with low PSA might have been diagnosed due to other clinical suspicions rather than screening). Also, the study relies on retrospective data, which inherently limits causality.

Minor Issues
• Some sentences are lengthy and could be restructured for better readability. For example, in the abstract:
“Low PSA concentrations were significantly associated with advanced disease and poor prognosis in all prostate cancer patients with GS 8–10, whereas they were a protective factor in those with GS 6–7, Stage T1 disease.”
→ Consider breaking this into two sentences for clarity.
• Ensure that figure legends and table captions provide sufficient context for standalone understanding.
• Check for consistency in figure formatting, including axis labels and font sizes.
• The manuscript mentions ethical approval, but it should also confirm whether all patients provided informed consent (or whether it was waived due to retrospective design).
• Clarify the abbreviation “LT” (local treatment) early in the manuscript, as it is used frequently.
• Consider engaging a professional language editor to refine the manuscript further.

·

Basic reporting

Introduction -Line34. Neuroendocrine differentiation of the tumor has to be accounted and this is well discussed in this letter I suggest to read and quote. Biganzoli EM et al. Benefit-harm ratio of the diagnostic workup in patients with prostate cancer of Gleason score from 9 to 10. Cancer 2021,

Methods: -The laboratory methods used for measuring PSA in both case series has to be considered. The methods aare not harmonized and may provide different results on the same samples.
This should be relevant to generalize the results. As an aid you can consider this paper by Professor Panteghini. Panteghini M. Verification of Harmonization of Serum Total and Free Prostate-Specific Antigen (PSA) Measurements and Implications for Medical
Decisions. Clin Chem. 2021;67:543-53

Discussion
-Line 139 Discussion
You discuss about the value of the single PSA result, but most of the studies currently use this value within nomograms to optimize the predicting value.
Ferraro, S.et al. Definition of Outcome-Based Prostate-Specific Antigen (PSA) Thresholds for Advanced Prostate Cancer Risk Prediction. Cancers 2021, 13, 3381
-Line 151 Discussion, I think that the neuroendocrine differentiation(see above) should be reconsidered here.

Experimental design

There are some flaws, PSA methods used in SEER and in the case series of your hospital should be considered

Validity of the findings

The finding are not a novelty and confirmed by the literature

Additional comments

None

---

## Round 0.2 · Major Revisions

The authors are requested to carefully revise the manuscript and answer the questions raised by the reviewers.

Reviewer 1 ·

Basic reporting

The authors have clearly laid out the background on prostate cancer.

Experimental design

This manuscript describes a retrospective analysis of existing data sets.

Validity of the findings

The use of SEER data may be subject to selection bias and other sources of confounding. The authors have compensated for this as much as possible and also adequately acknowledged this limitation in the data. The advantage of the SEER data is its size and coverage of a wide range of degrees of disease severity.

Additional comments

The authors have responded adequately to the comments of the reviewers of the original version of this manuscript.

Reviewer 2 ·

Basic reporting

no comment

Experimental design

no comment

Validity of the findings

no comment

Additional comments

This manuscript presents a comprehensive retrospective analysis using SEER data to assess the prognostic implications of low PSA (<4 ng/mL) versus intermediate PSA (4–10 ng/mL) in patients with prostate cancer, with stratification by T stage and Gleason score (GS). The authors further develop and externally validate a prognostic nomogram for high-risk patients (GS 8–10) with low PSA values. The topic is clinically relevant and addresses a nuanced aspect of prostate cancer risk stratification, especially given the non-linear and sometimes paradoxical relationship between PSA levels and tumor aggressiveness.
Major Issues
• While the nomogram development and validation are statistically sound, the clinical implications of low PSA in the high-risk group require more nuanced discussion, particularly regarding how this tool could alter clinical decision-making in practice.
• The nomogram was constructed only for the high-risk cohort (GS 8–10), which is justifiable but limits its utility for broader populations. The rationale for not building a model for the full cohort (or even including GS 6–7 T3–T4 patients) should be clarified.
• Although the discussion touches upon neuroendocrine differentiation and PSA-inactive tumors, the lack of genetic or biological marker analysis (beyond speculation) is a limitation, particularly when explaining mechanisms behind low PSA, aggressive tumors.
• The external validation cohort from Qingdao is relatively small (n = 199) and racially homogeneous. The authors acknowledge this limitation but could further elaborate on its implications for generalizability, especially in diverse populations.
• Details regarding treatment (especially use and duration of ADT) are missing from the SEER data. Given that ADT is a cornerstone of prostate cancer therapy, this limits the ability to fully contextualize survival outcomes. These limitations should be discussed more explicitly in both the Methods and Discussion sections.
Minor Issues
• Minor English grammar and stylistic issues exist throughout the manuscript (e.g., "f 4 ng/mL" instead of "≤ 4 ng/mL" or "<4 ng/mL"). Proofreading by a native English speaker or editing service is recommended.
• Terms like “protective factor” vs “risk factor” could be better defined statistically and clinically.
• Figure quality is adequate but could be enhanced. Nomogram (Figure 2) should be reproduced at higher resolution.
• The manuscript includes occasional typographical artifacts (e.g., “f 10” or “GS g8”) that should be corrected.

---

## Round 0.3 · Minor Revisions

The authors are requested to carefully revise the manuscript and answer the questions raised by the reviewers.

**Language Note:** The review process has identified that the English language must be improved. PeerJ can provide language editing services - please contact us at [email protected] for pricing (be sure to provide your manuscript number and title). Alternatively, you should make your own arrangements to improve the language quality and provide details in your response letter. – PeerJ Staff

Reviewer 2 ·

Basic reporting

The manuscript is written in generally clear and professional English. There are occasional grammatical errors and awkward phrasings that could benefit from professional copyediting.

Experimental design

The study uses a robust retrospective design analyzing SEER data (n=233,554) and externally validates findings on a Chinese institutional cohort (n=199). Stratification by both GS and T stage before comparing low (≤4 ng/mL) and intermediate PSA (4.1–10 ng/mL) is an appropriate and novel approach.

Validity of the findings

The data convincingly show that low PSA levels are associated with poor prognosis in high-GS tumors but may be protective in low-GS, T1 tumors. The high-risk model is statistically sound with good AUCs (training AUCs: 0.735–0.788; external validation: up to 0.955). Internal calibration and DCA support the model’s utility. The external cohort adds strength, despite its smaller size and single-center nature.

Additional comments

The study is novel, well-executed, and adds clinically meaningful insight into PSA interpretation. With minor revisions to clarify terminology, elaborate on subgroup definitions, and polish the manuscript’s language, it will be suitable for publication.

---

## Round 0.4 · accepted · Accept

After revisions, one reviewer agreed to publish the manuscript. There is one reviewer left with a minor revision, and I think the author has responded adequately. I also reviewed the manuscript and found no obvious risks to publication. Therefore, I also approved the publication of this manuscript.